# Predation Capacity of the Banded Thrips *Aeolothrips intermedius* for the Biological Control of the Onion Thrips *Thrips tabaci*

**DOI:** 10.3390/insects13080702

**Published:** 2022-08-04

**Authors:** Linda Abenaim, Stefano Bedini, Alessio Greco, Paolo Giannotti, Barbara Conti

**Affiliations:** Department of Agriculture, Food and Environment, University of Pisa, Via del Borghetto 80, 56126 Pisa, Italy

**Keywords:** banded thrips, onion thrips, predation time, number of prey

## Abstract

**Simple Summary:**

*Aeolothrips intermedius* is a native predator of *Thrips tabaci*, one of the most harmful thrips for many open field crops. Our study evaluated the predatory efficiency of *A. intermedius* against *T. tabaci* adults. The results showed that *A. intermedius* can be considered a promising resource for the biological control of onion thrips.

**Abstract:**

The onion thrips, *Thrips tabaci*, is a main insect pest for many field crops worldwide, with a particular preference for the species of the genus *Allium*. *Aeolothrips intermedius* is a banded thrips, whose larvae are considered the primary native predator of *T. tabaci*. Due of their predatory behaviour, *A. intermedius* larvae are considered a good candidate for biological control against thrips pests. However, limited information is available on the specific predation rate of *A. intermedius* against *T. tabaci*. The aim of our study was to evaluate the predatory efficiency of *A. intermedius* larvae against *T. tabaci* adults. Predation assays, performed under laboratory conditions, indicated that *A. intermedius* larvae begin to prey after an average of about 23 min, and the time taken by an *A. intermedius* larva to capture and subdue the prey until its death is about 26 min. Furthermore, the maximum number of prey that the *A. intermedius* larvae are able to kill in 12 h is up to eight adults of *T. tabaci/A. intermedius* larva.

## 1. Introduction

The onion thrips *Thrips tabaci* Lindeman, 1889 (Thysanoptera: Thripidae) (Figure 1), is one of the most harmful polyphagous pests for many field crops [1]. In particular, *T. tabaci* causes huge damage to Alliaceae [2] and Brassicaceae crops [3]. In onion crops, *T. tabaci*, if uncontrolled, can reduce the yield by as much as 75% [4]. In addition, *T. tabaci* is also a vector of the Iris Yellow Spot Virus (IYSV), a significant disease affecting onion, leek, iris, and wild *Allium* species [5].

The use of synthetic insecticides is the most adopted method for the control of this pest. However, because of its high reproductive rate, short generation time, and wide host range [6], *T. tabaci* is difficult to control and the treatments have to be repeated frequently, which results in the presence of insecticide residues in food products, resistance development, and high costs [7,8,9,10,11]. Due to the increasing concern about the side effects of the overuse of synthetic insecticides, new eco-friendly approaches for managing thrips pests, such as biological controls, are demanded by growers and consumers. In this scenario, biological control systems may help to reduce both pest populations and virus incidence not only in conventional and IPM farming, but also in organic farming. Among the biological control systems, some natural products [12,13,14] seem useful for harmful thrips control, even if they are not decisive [15].

Among the *T. tabaci* biological control agents, some predators, such as predatory mites (*Neoseiulus* spp.) and Hemiptera (*Orius* spp.), have been reported [16]. However, the efficacy of these predators is not considered decisive [6]. Among the *T. tabaci* predators, *Aeolothrips intermedius* Bagnall, 1934 (Thysanoptera: Aeolothripidae), is a cosmopolitan species [17,18], also known as “banded thrips” (Figure 2), even if all the congeneric species present the same characteristic of black and white banded wings. Both adults and larvae are floricolous (mainly found in Leguminosae, Rosaceae, and Poaceae; Table 1) and while adults feed mainly on flowers and pollen by their stem-sucking-rasping mouthparts, *A. intermedius* larvae prey on more than 44 species of Thysanoptera [19].

In Europe, *A. intermedius* larvae are reported as predators of *Heliothrips haemorrhoidalis* Bouché, *Odontothrips confusus* Priesner, and some species belonging to the genus *Haplothrips* [20]. *A. intermedius* is considered the primary native predator of *Thrips tabaci* [20,21,22,23,24] and a good candidate for the biological control of onion thrips in field conditions [25]. However, despite that *A. intermedius* has been reported as a very effective predator that may play an important role in the biological control of Thysanoptera [26], there is little information on its specific predation rate against *T. tabaci*. Thus, the aim of our study was to assess, using laboratory tests, the predatory efficiency of *A. intermedius* larvae against the crop insect pest *T. tabaci.* In particular, the predation tests carried out in this study were aimed to evaluate (a) the predation times of *A. intermedius* larvae on *T. tabaci* adults; (b) the predation rate of *A. intermedius* larvae on *T. tabaci* adults.

**Table 1 insects-13-00702-t001:** Plant species for which the predatory thrips *Aeolothrips intermedius* have been recorded.

Plant Species	Family	References
*Albizzia juliprissin*	Leguminosae	Marullo, 1991 [27]
*Allium cepa*	Alliaceae	Trdan, 2005 [20]
*Avena sativa*	Poaceae	Trdan, 2005 [20]
*Brassica* sp.	Brassicaceae	Marullo, 1993 [22]
*Coronilla varia*	Leguminosae	Marullo, 1993 [22]
*Dianthus* sp.	Cariophillaceae	Trdan, 2005 [20]
*Diplotaxis* sp.	Brassicaceae	Marullo, 1993 [22]
*Echium vulgare*	Boraginaceae	Zur Strassen, 1987 [28]
*Galeca officinalis*	Leguminosae	Zur Strassen, 1987 [28]
*Galeopsis ladanum*	Lamiaceae	Zur Strassen, 1991 [29]
*Gallium* sp.	Rubiaceae	Zur Strassen, 1991 [29]
*Genista sagittalis*	Leguminosae	Zur Strassen, 1991 [29]
*Helianthus annuus*	Asteraceae	Trdan, 2005 [20]
*Helianthemum nummularium*	Cistaceae	Zur Strassen, 1991 [29]
*Hordeum vulgare*	Poaceae	Trdan, 2005 [20]
*Jasione montana*	Campanulaceae	Zur Strassen, 1991 [29]
*Lactuca sativa*	Cichoriaceae	Trdan, 2005 [20]
*Malus communis*	Rosaceae	Marullo, 1993 [22]
*Medicago sativa*	Leguminosae	Mound, 1968 [30]
*Melilotus* sp.	Leguminosae	Marullo, 1993 [22]
*Nicotiana tabacum*	Solanaceae	Trdan, 2005 [20]
*Pisum sativum*	Leguminosae	Trdan, 2005 [20]
*Pteridium aquilinum*	Hypolepidaceae	Marullo, 1991 [27]
*Quercus pubescens*	Fagaceae	Marullo, 1991 [27]
*Sinapis* sp.	Brassicaceae	Zur Strassen, 1991 [29]
*Trifolium pratense*	Leguminosae	Trdan, 2005 [20]
*Trifolium repens*	Leguminosae	Trdan, 2005 [20]
*Triticum aestivum*	Poaceae	Trdan, 2005 [20]
*Zea mays*	Poaceae	Trdan, 2005 [20]

## 2. Materials and Methods

### 2.1. Insect Specimens

*A. intermedius* and *T. tabaci* were reared in four alfalfa (*Medicago sativa* L.) square parcels (1 m × 1 m) and covered with a screen (0.6–0.8 mm mesh size) at the Centro Enrico Avanzi of the University of Pisa (CiRAA) (Pisa, Italy) during the period May–September 2021. The two species were collected daily from alfalfa flowers, present in the parcels from July until the end of September, to be tested. The alfalfa flowers collected were shacked on white paper to allow for an easy visual inspection of the larval and adult *A. intermedius* and *T. tabaci* populations. One *A. intermedius* larva (predator) and one *T. tabaci* adult (prey) were collected from the surface with a fine-tipped wet brush and were transferred into a Huffaker cage [31] (Figure 3) with a green linden (*Tilia platyphyllos*) leaf as vegetal support to allow for observations during the predation tests [24].

The Huffaker cages were maintained under laboratory conditions at 25 ± 2 °C, 65 ± 5% RH, and 16:8 = L:D photoperiod. Since contamination of the species in the rearing parcels could occur, to ensure the correct identification of both the prey and the predator, some randomly chosen specimens were collected and mounted onto slides by the current methodology [32], which were then observed under the microscope for a specific identification confirmation. In particular, the predator larvae were fed until they produced cocoons and the adults emerged (Figure 4).

### 2.2. Predation Time

To evaluate the time spent by the predator to attack and kill the prey, the *A. intermedius* larvae, collected the day before the test, were starved for 24 h. On the day of the test, an *A. intermedius* larva (L-II) and a *T. tabaci* adult were put in a Huffaker cage and the time when the predation started, the time spent to capture and subdue the prey, and the final time when the predator stops sucking the prey were recorded (Figure 5). In the rare case that a predation did not occur, the test was discarded. The test was replicated until 30 valid predations occurred.

### 2.3. Predation Rate

To determine the predation rate of the *A. intermedius* larvae on the *T. tabaci* adults, the number of prey killed by each predator (prey/predator) over 12 h was assessed. The predators were collected the day before the test and were starved for 24 h. For each replica, one *A. intermedius* larva and three *T. tabaci* adults were put in a Huffaker cage. The predation was recorded every three hours from 9.00 AM until 9.00 PM (controls were performed at 12 AM, 3 PM, 6 PM, and 9 PM). If at the check time the predator had killed all prey, three more prey were added. The number of prey inserted in the Huffaker cage at each check, the number of prey killed, and the percentage of prey killed over 12 h were recorded. The test was replicated until 30 valid predations occurred. Three *T. tabaci* were put in a Huffaker cage as a control for each test and the eventual natural mortality was checked every three hours from 9.00 AM until 9.00 PM.

## 3. Results

The predation time bioassay results indicated that the *A. intermedius* larvae begin to prey on the *T. tabaci* adults after about 23 ±14 min, and the time spent by the *A. intermedius* larvae in capturing and subduing the prey until it leaves the victim dead is about 26 ± 12 min. The total time from the first visual contact of the predator with the prey until the end of predation (when the predator leaves the prey) is about 49 ± 18 min.

The predation rate tests showed that the maximum number of *T. tabaci* adults that the *A. intermedius* larvae were able to kill over a period of 12 h was eight prey/predator (88.89%). The average quantity of prey killed by each predator over 12 h was 3.77 ± 0.15 (Table 2), with a mode of two prey/predator (Figure 6). No natural mortality was recorded in the control tests.

## 4. Discussion

*Thrips tabaci* is a problematic pest, whose control is hampered by a high reproductive rate of the species, a short generation time, and a wide range of hosts [6]. Biological controls by predatory Thysanoptera [25,33,34] showed promising results. In particular, *A. intermedius* has been tested to determine the correct timing for chemical treatments, the factors affecting the dynamics of its interaction with the *T. tabaci* population, its susceptibility to commonly used insecticides, and its interaction with other predators [25,33,34]. However, key information on the biological and ecological aspects of the predation of *A. intermedius* on onion thrips is still lacking [20,33,34], and to our knowledge, this is the first time that the predatory efficacy of *A. intermedius* against *T. tabaci* has been assessed.

Our results are in line with the predatory efficacy reported by Fathi et al. [33] for *Orius niger* (Hemiptera: Anthocoridae) and *A. intermedius* against *T. tabaci*. The authors observed high predation rates with an additive effect when the two predators were used together, while the predation was lower when each predator was used separately as follows: 77% for the *A. intermedius* predation and 91% for the *O. niger* predation [33]. Such percentages are very close to the 88.9% (eight prey killed over the nine administrated) obtained as the maximum predation rate in our tests. Similarly, Deligeorgidis [35] observed that, at a ten prey density, one *O. niger* female was able to prey on 73.63% of the *T. tabaci* and 59.66% of the *F. occidentalis* over 24 h. However, a lower predation efficacy was observed for *Coccinella septempunctata* (Coleoptera Coccinellidae) with about 51.37% of the preyed *T. tabaci* adults [36].

The predation rates observed in this experiment for the second instar *A. intermedius* larvae were close to those of the same instar of other Aeolothripidae predators currently commercialized. In fact, Mahendran and Radhakrishnan [37] observed that the commercialized Aeolothripidae predator *Franklinothrips vespiformis* larvae are able to prey on 12 *Scirtothrips bispinosus* (a Thripidae harmful for tea plantations) adults within 24 h. In our experiment, the prey (*T. tabaci* adults) were administrated sequentially (three prey every three hours) in order to keep the prey/predator rate quite constant during the experiment. However, Khamis and Jabbar [38] and El-Sheikh et al. [39] observed that the number of prey (*T. tabaci* adults) killed by *Orius albipennis* was significantly correlated with the prey density varying from 10 to 25 prey per day for prey densities from 10 to 60 per predator. This indicates that under field working conditions with higher prey densities, *A. intermedius* could increase its predatory efficacy and be a very effective predator for the control of *T. tabaci*, with predation rates in line with other predators already commercialized (*F. vespiformis*, *Orius* sp., and predatory mites).

As expected, due to the difference in the size of the mites compared to the predatory Thysanoptera, the predation rates obtained in this experiment are higher than those reported by other authors for mites. This could be due to due to the difference in size between the mites and the Thysanoptera. Actually, Messelink et al. [40] observed that the predator mite *Amblyseius swirskii* (Athias-Henriot) is able to prey on about six *Frankliniella occidentalis* adults in 48 h, while Berndt et al. [41] reported predation rates of 3.5 (±0.5) and 1.64 (±0.3) *F. occidentalis* per day by *Hypoaspis aculeifer* females and *Stratiolaelaps miles* (Berlese) females, respectively. Lower predation rates than the one we observed for *A. intermedius* were obtained by Walzer et al. [42] also for the mite *Neoseiulus californicus*, which preyed on an average of 3.19 *T. tabaci* larvae per day.

A main drawback to the widespread use of *A. intermedius* could be the difficulty of rearing. In fact, even if Bournier et al. [21] published a complete breading protocol for *A. intermedius,* other authors reported the difficulties in rearing the species under laboratory conditions [43]. In this work, we obtained *A. intermedius* larvae from spontaneous breeding in open fields. Under such conditions, it was quite easy to obtain samples for the biological assays. In this regard, in previous observations in the open field, Conti et al. [44] reported three peaks in the presence of an *A. intermedius* adult as follows: the first in early June, the second in mid-July, and the third in early August. This is in accordance with the results of Orosz et al. [25] and Trdan et al. [45], who recorded the first and second adult peaks of *A. intermedius* in the same period (except for August, probably due to the different climatic conditions between central Italy and Hungary). Overall, these results support the possibility of obtaining large numbers of *A. intermedius* larvae for use in the biological control of *Thrips tabaci* without a complicated breeding procedure.

## 5. Conclusions

In conclusion, the present study allowed us to describe the *T. tabaci*–*A. intermedius* prey/predator interaction. The results demonstrated that *A. intermedius* could be considered a promising resource for the biological control of onion thrips in fields. Overall, our data show that *A. intermedius* can contribute to reducing the population of *T. tabaci*. In addition, since *A. intermedius* has been observed in over 40 vegetal species as a predator of 18 thrips species, especially those belonging to the suborder of Terebrantia, such as the onion thrips [19], it may represent a very versatile biocontrol agent that is able to adapt to different plant hosts and a wide range of harmful thrips.

## Figures and Tables

**Figure 1 insects-13-00702-f001:**
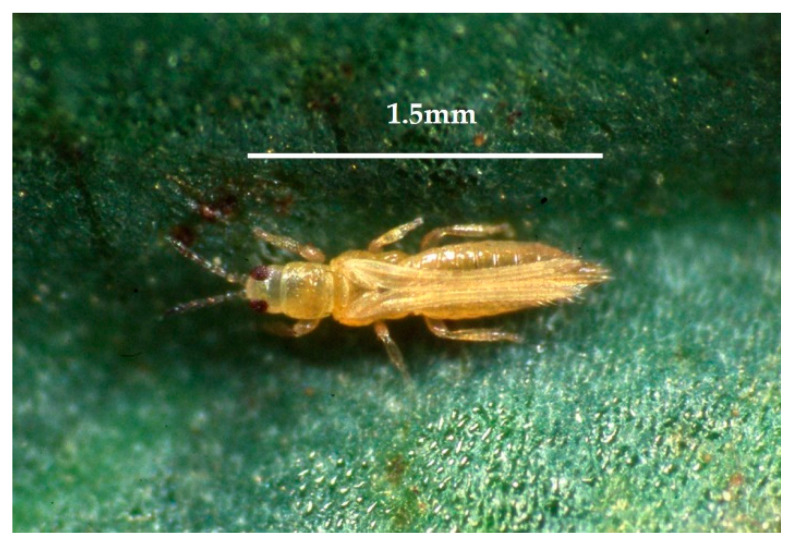
Adult of the onion thrips *Thrips tabaci* (Photo by R. Antonelli).

**Figure 2 insects-13-00702-f002:**
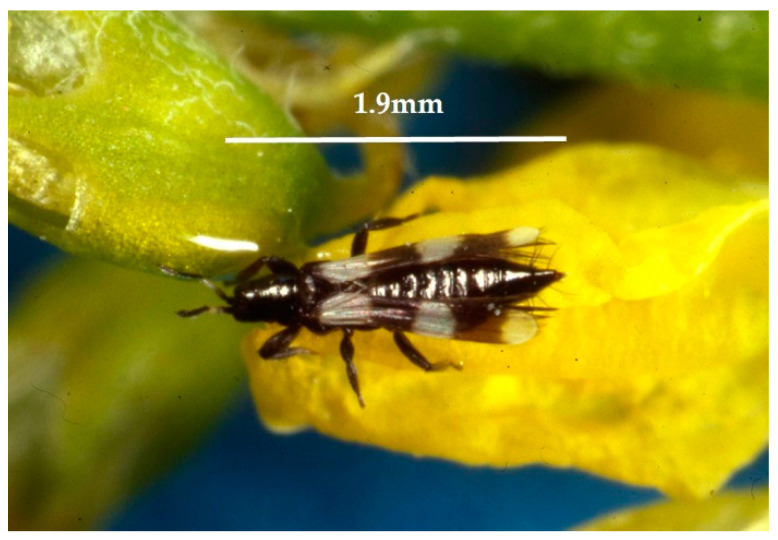
Female of the predatory thrips *Aeolothrips intermedius* (Photo by R. Antonelli).

**Figure 3 insects-13-00702-f003:**
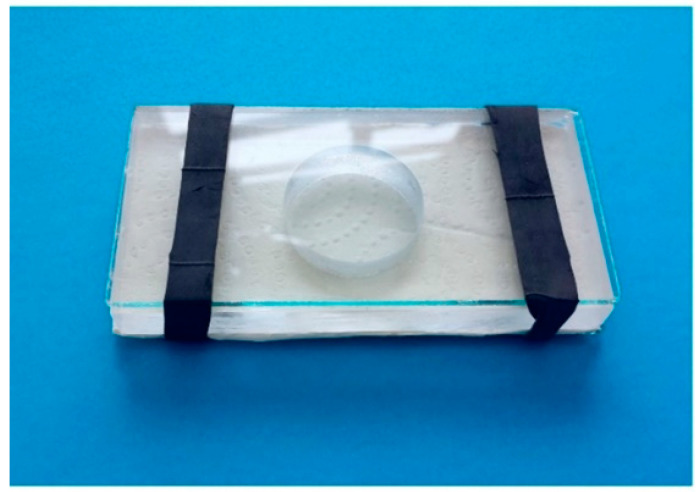
Huffaker cage used in the experiments (Photo by R. Antonelli).

**Figure 4 insects-13-00702-f004:**
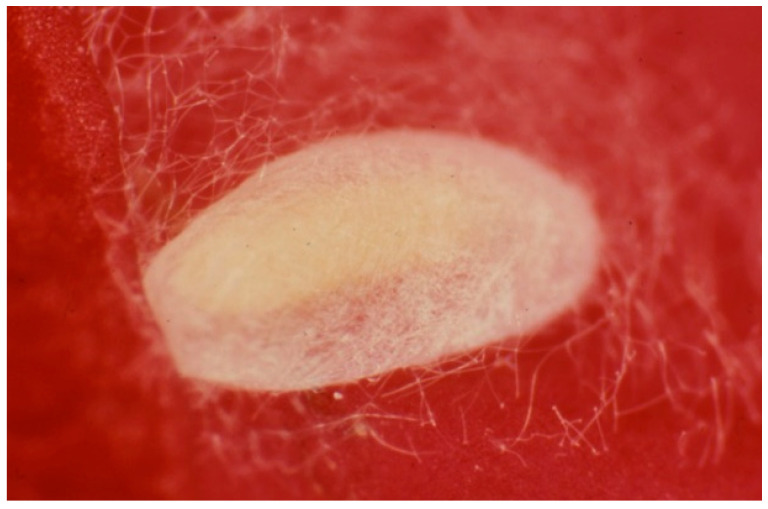
Cocoon of the predatory thrips *Aeolothrips intermedius* (Photo by R. Antonelli).

**Figure 5 insects-13-00702-f005:**
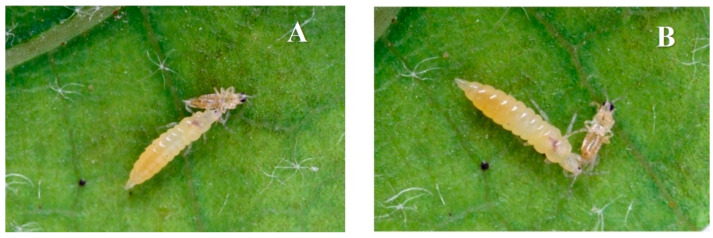
Predation of an *Aeolothrips intermedius* larva on an onion thrips *Thrips tabaci* adult. (**A**) start of the predation (the predator captures and subdues the prey); (**B**) end of the predation (the predator stops feeding on the prey).

**Figure 6 insects-13-00702-f006:**
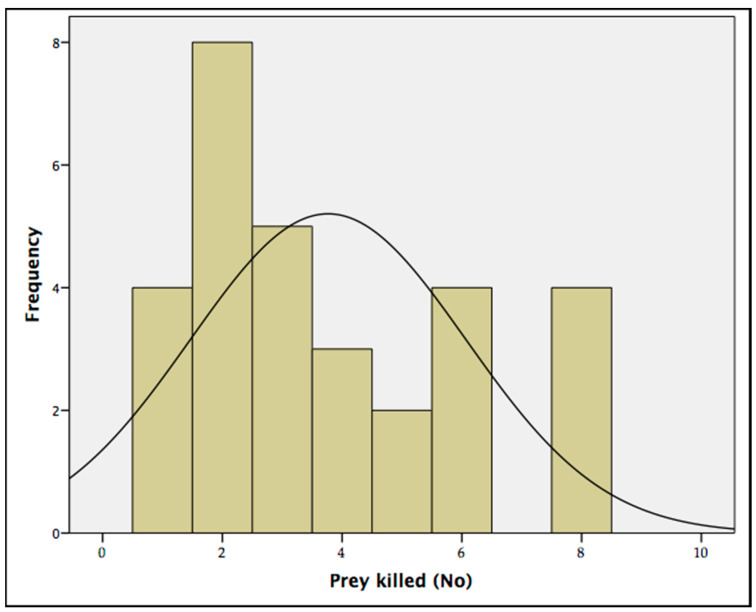
Mode of the predation rate of the predator larvae *Aeolothrips intermedius* on onion pest *Thrips tabaci* adults. The histogram represents the frequencies of *Thrips tabaci* preyed over 12 h. The curve represents a standard normal distribution.

**Table 2 insects-13-00702-t002:** Predation rate of *Aeolothrips intermedius* larvae on the onion pest *Thrips tabaci* adults.

Time (h)	N. of Prey Administrated ^a^	N. of Prey Killed ^b^
3	3	1.37 ± 0.18
6	6	2.03 ± 0.14
9	9	2.80 ± 0.16
12	9	3.77 ± 0.15

^a^ Total number of *Thrips tabaci* adults administrated to an *Aeolothrips intermedius* predatory larva; ^b^ mean number (± standard error) of the *T. tabaci* adults killed by an *A. intermedius* larva.

## Data Availability

The data presented in this study are available on request from the corresponding author.

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
