# Peer review of "Predation Capacity of the Banded Thrips Aeolothrips intermedius for the Biological Control of the Onion Thrips Thrips tabaci"

_insects, 2022, doi:10.3390/insects13080702_

Round 1

Reviewer 1 Report

In the paper submitted by Abenaim et al. the potential role of the banded trips as an agent for biological control is investigated. I think the paper is interesting as is within the scope the the journal insects.

I have a few comments about the paper:

Introduction:

line 28 year of Lindemans description missing.

line 35 Photo by whom?

line 42: biological control is also important for organic farming, maybe you want to add this.

line 45: is this general about biological control, then I miss wasps, nematodes and other taxa. Or is it for trips only. please clarify.

line 51 mouthparts are on Fig 2 not visible.

MM:

Did you think of a control? could the trips also die naturally without being parasitized?

separate and name the two parts of the experiment: predation time and predation rate

Results and discussion

In general I don’t like if these two sections are together and this example is very good, as line 111-128 are discussion and later the results start.

I am also confused with the average percentage of predation (75,9) or the predation rate (3,8), is this the same data or how is it used?

Also the comparison with the literature are confusing as the times of observations are not the same (48 h and 12 h, line 150 and 151).

What is the curve in Fig 6.

line 176-177: these are interesting observations in the field, but this is part of the results and should be included in the MM section.

maybe a table with the species of plants, their pests (trips) and their predators could add important information’s and you could show, where your results are relevant (line 188).

the references need to be checked again and adapted to the insects journal.

I am not sure what reference 18 and 19 is? Is this a collection or a paper?

Author Response

Introduction:

line 28 year of Lindemans description missing.

R: We added the date of the description of Thrips tabaci

line 35 Photo by whom?

R: We named the Authors of the photos in Figure 1-4 and we added the acknowledgements at the end of the text.

line 42: biological control is also important for organic farming, maybe you want to add this.

R: Thanks for the suggestion, we added organic farming to the sentence as suggested.

line 45: is this general about biological control, then I miss wasps, nematodes and other taxa. Or is it for trips only. please clarify.

R: We specify that we mean for Thrips tabaci biological control.

line 51 mouthparts are on Fig 2 not visible.

R: We have moved the citation of the Fig. 2 in a more appropriate place.

MM:

Did you think of a control? could the trips also die naturally without being parasitized?

R: For the first experiment (Predation time) no control were necessary since we considered the test as successful only if the predation (of the live prey) occurs. For the second experiments (predation rate) the control tests were performed (lines 118-119) but since no control thrips died we did not apply any correction to the mortality rates.

separate and name the two parts of the experiment: predation time and predation rate

R: We separated the two parts of the experiment as suggested

Results and discussion

In general I don’t like if these two sections are together and this example is very good, as line 111-128 are discussion and later the results start.

R: In general, we agree with the Reviewer. However, when data is concise as well as the discussion, it may be better to put them together. “Insects” does not include short communications in the types of the publication formats but, since it allows the authors to be free to choice the format they prefer we would prefer to leave a single section for “Results and Discussion”.

I am also confused with the average percentage of predation (75,9) or the predation rate (3,8), is this the same data or how is it used?

R: We now reported the data as predation rate (number of preys killed by the predator over 12 hours) only and modified Table 2 accordingly. We thank the reviewer for the suggestion.

Also the comparison with the literature are confusing as the times of observations are not the same (48 h and 12 h, line 150 and 151).

R: Yes, the Reviewer is right. Unfortunately, the time of observation are not the same. For a better comparison of the results we added a conversion of the rate to 12 h (line 165-166 without track revisions).

What is the curve in Fig 6.

R: thanks for the suggestion, we added the information in the figure 6 caption.

line 176-177: these are interesting observations in the field, but this is part of the results and should be included in the MM section.

R: We apologize for the misunderstanding since the open field observations are referred to a previous work. We better specify it in the text.

maybe a table with the species of plants, their pests (trips) and their predators could add important information’s and you could show, where your results are relevant (line 188).

R: We thank the Reviewer for the suggestion. We added the table as “table 1”

the references need to be checked again and adapted to the insects journal.

R. We checked and corrected all the references

I am not sure what reference 18 and 19 is? Is this a collection or a paper?

R: We have moved the citations 18 and 19 (now 17 and 18, two faunistic investigations) in the introduction (lines 50-51 without track revisions).

Reviewer 2 Report

This is well designed study that produced interesting results.  One minor suggestion is to add a transition sentence at line 147 to introduce the idea of mite predation of thrips pests. 

Author Response

One minor suggestion is to add a transition sentence at line 147 to introduce the idea of mite predation of thrips pests. 

R: We introduced the idea of mite predation of thrips pests at lines 126-127 (without track revisions).

Round 2

Reviewer 1 Report

The paper was improoved and is ready for publications soon.

I am still a bit confused with line 154 and the prey/larva, maybe use only on term as it is confunsing: I seems you calculate prey PER larva but you want to present prey / predator or larva / predator right?

Is the plural from prey preys?

Author Response

We thank very much the reviewers and the editor for their expert and accurate advice and suggestions which greatly improved the manuscript.

Best Regards,

Stefano Bedini

Replies to Reviewer 1

I am still a bit confused with line 154 and the prey/larva, maybe use only on term as it is confunsing: I seems you calculate prey PER larva but you want to present prey / predator or larva / predator right?

R: We agree with the reviewer. Although the term prey/larva should be correct as we have specified in M&M that A. intermedius larvae are the predator and the adults of T. tabaci the prey, “A larva of A. intermedius (predator) and an adult of T. tabaci (prey) ”(line 80), the term could be easily misunderstood. For this, we have changed throughout the text "prey/larva" with "prey/predator"

Is the plural from prey preys?

We apologize for the error. We changed "preys" to "prey" throughout the text